# Your Classifier Can Be Secretly a Likelihood-Based OOD Detector

**Jirayu Burapacheep**                                                *jirayu@stanford.edu*
*Department of Computer Science*
*Stanford University*

**Yixuan Li**                                                         *sharonli@cs.wisc.edu*
*Department of Computer Sciences*
*University of Wisconsin - Madison*

**Reviewed on OpenReview:** *https://openreview.net/forum?id=FmA1JPWBM8*

## Abstract

The ability to detect out-of-distribution (OOD) inputs is critical to guarantee the reliability of classification models deployed in an open environment. A fundamental challenge in OOD detection is that a discriminative classifier is typically trained to estimate the posterior probability $p(y|\mathbf{z})$ for class $y$ given an input $\mathbf{z}$, but lacks the explicit likelihood estimation of $p(\mathbf{z})$ ideally needed for OOD detection. While numerous OOD scoring functions have been proposed for classification models, these estimate scores are often heuristic-driven and cannot be rigorously interpreted as likelihood. To bridge the gap, we propose Intrinsic Likelihood (**INK**), which offers rigorous likelihood interpretation to modern discriminative-based classifiers. Specifically, our proposed INK score operates on the constrained latent embeddings of a discriminative classifier, which are modeled as a mixture of hyperspherical embeddings with constant norm. We draw a novel connection between the hyperspherical distribution and the intrinsic likelihood, which can be effectively optimized in modern neural networks. Extensive experiments on the OpenOOD benchmark empirically demonstrate that INK establishes a new state-of-the-art in a variety of OOD detection setups, including both far-OOD and near-OOD.

## 1 Introduction

The problem of out-of-distribution (OOD) detection has garnered significant attention in reliable machine learning. OOD detection refers to the task of identifying test samples that do not belong to the same distribution as the training data, which can occur due to various factors such as the emergence of unseen classes in the model's operating environment. The ability to detect OOD samples is crucial for ensuring the reliability and safety of machine learning models (Amodei et al., 2016), especially in applications where the consequences of misidentifying OOD samples can be severe. To facilitate the separation between in-distribution (ID) and OOD data, a rich line of works (Yang et al., 2021b) have developed OOD indicator functions, which, at the core, compute a scalar score for each input signifying the degree of OOD-ness. The design of these scoring functions has emerged as a critical cornerstone in achieving effective OOD detection.

By definition, OOD data inherently diverges from ID data by means of their data density distributions, rendering likelihood or log-likelihood an ideal scoring function for detection. However, a fundamental dilemma in OOD detection is that a discriminative classifier is typically trained to estimate the posterior probability $p(y|\mathbf{z})$ for class $y$ given an input $\mathbf{z}$, but lacks the explicit likelihood estimation of $p(\mathbf{z})$ needed for OOD detection. While many OOD scoring functions have been proposed for classification models, they are either heuristic-driven and cannot be rigorously interpreted as log-likelihood, or impose strong assumption on the density function that can fail to hold in test time (Lee et al., 2018b). Against this backdrop, the key research question of this paper thus centers on:

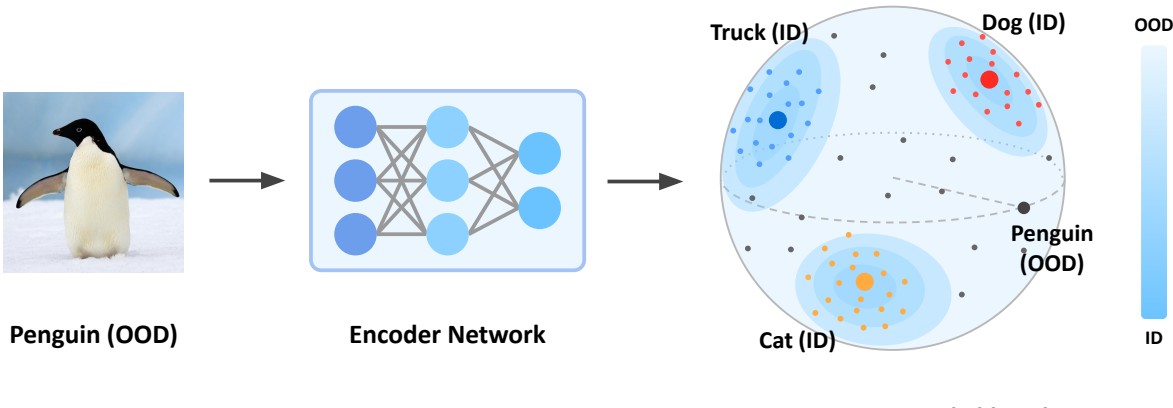

Figure 1: Overview of intrinsic likelihood framework for OOD detection. The neural network is trained on the in-distribution (ID) data, which lies on the unit hypersphere in the latent space. The intrinsic likelihood score is theoretically equivalent to the log-likelihood $\log p(\mathbf{z})$, which suits OOD detection.

*RQ: How to design an OOD scoring function that can offer rigorous likelihood interpretation to modern discriminative-based classifiers without resorting to separate generative models?*

To bridge the gap, we introduce a novel framework **IN**trinsic li**K**elihood (**INK**) for classifier-based OOD detection, which can be theoretically reasoned from a log-likelihood perspective. The key insight is that we need to incorporate explicit probabilistic modeling of the data distribution $p(\mathbf{z})$ directly into optimizing the posterior $p(y|\mathbf{z})$, which is effectively achieved by our framework. Specifically, our proposed INK score operates on the constrained latent embeddings of a discriminative classifier, which are modeled as a mixture of hyperspherical embeddings with constant norm (see Figure 1). Under our probabilistic model, each class has a mean vector in the hypersphere, where the class-conditional density function decays from the center exponentially by the similarity between an embedding to the center. The proposed INK score is formalized by drawing a novel connection between the hyperspherical distribution and the density function under our probabilistic model. Theoretically, we show that the INK score is equivalent to the log-likelihood score $\log p(\mathbf{z})$ (up to some constant difference), and thus can act rigorously as a density-based score for OOD detection. Though simple and elegant in hindsight, establishing this connection was non-trivial. We lay out the full theoretical derivation in Section 3.1, along with how to effectively optimize for the intrinsic likelihood in the context of discriminative neural networks in Section 3.2.

We show that our theoretical guarantee indeed translates into competitive empirical performance (Section 4). We extensively evaluate our method on the latest OpenOOD benchmarks (Zhang et al., 2023a), containing CIFAR and ImageNet-1k as ID datasets. Our method exhibits significant performance improvements when compared to the state-of-the-art method ASH (Djurisic et al., 2023), which employs a heuristic-driven activation shaping approach and lacks a rigorous likelihood interpretation. Moreover, we show that our method is effective under both far-OOD and near-OOD scenarios and generalizes to different model architectures, including ViT (Dosovitskiy et al., 2021). The overall evaluation demonstrates notable strengths of our approach in both empirical performance and theoretical underpinnings.

## 2 Preliminaries

Let $\mathcal{X}$ and $\mathcal{Y} = \{1, \ldots, C\}$ represent the input space and ID label space, respectively. The joint ID distribution, represented as $P_{X_I Y_I}$, is a joint distribution defined over $\mathcal{X} \times \mathcal{Y}$. During testing time, there are some unknown OOD joint distributions $D_{X_O Y_O}$ defined over $\mathcal{X} \times \mathcal{Y}^c$, where $\mathcal{Y}^c$ is the complementary set of $\mathcal{Y}$. We also denote $p(\mathbf{x})$ as the density of the ID marginal distribution $P_{X_I}$. According to Fang et al. (2022), OOD detection can be formally defined as follows:

**Definition 2.1** (**OOD Detection**). *Given labeled ID data $\mathcal{D}_{\text{in}} = \{(\mathbf{x}_1, y_1), ..., (\mathbf{x}_N, y_N)\}$, which is drawn from $P_{X_1 Y_1}$ independent and identically distributed, the aim of OOD detection is to learn a predictor $g$ by using $\mathcal{D}_{\text{in}}$ such that for any test data $\mathbf{x}$: 1) if $\mathbf{x}$ is drawn from $P_{X_1}$, then $g$ can classify $\mathbf{x}$ into correct ID classes, and 2) if $\mathbf{x}$ is drawn from $P_{X_O}$, then $g$ can detect $\mathbf{x}$ as OOD data.*

**Definition 2.2** (**Likelihood-based OOD Detector**). *Given ID data density function $p_{\boldsymbol{\theta}}(\cdot)$ estimated by a model $f_{\boldsymbol{\theta}}$ and a pre-defined threshold $\lambda$, then for any data $\mathbf{x} \in \mathcal{X}$,*

$$g(\mathbf{x}) = ID, \ \ if \ p_{\boldsymbol{\theta}}(\mathbf{x}) \geq \lambda; \ \ otherwise, \ g(\mathbf{x}) = OOD. \tag{1}$$

The performance of OOD detection thus heavily relies on the estimation of data density $p_{\boldsymbol{\theta}}(\mathbf{x})$. *However, in supervised learning, the classifiers directly estimate the posterior probability $p_{\boldsymbol{\theta}}(y|\mathbf{x})$ instead of the likelihood $p_{\boldsymbol{\theta}}(\mathbf{x})$, which makes likelihood-based OOD detection non-trivial.* Our focus on discriminative models differs from unsupervised OOD detection using generative-based models, which do not offer classification capability. For completeness, we review these studies extensively in the related work (Section 5).

## 3 Methodology

In this section, we introduce our proposed framework, **IN**trinsic li**K**elihood (INK), for OOD detection. We first provide an overview of the key challenge and motivation. Then, we introduce the notion of intrinsic likelihood and theoretically justify its feasibility as an indicator function of OOD in Section 3.1. Finally, in Section 3.2, we discuss how to optimize neural networks to achieve the desired intrinsic likelihood for OOD detection.

A classifier is typically trained via maximum likelihood estimation (MLE) on the training dataset $\mathcal{D}_{\text{in}} = \{(\mathbf{x}_i, y_i)\}_{i=1}^N$, in order to maximize the predictive probability for the corresponding class:

$$\text{argmax}_{\theta} \prod_{i=1}^N p(y = y_i \mid \mathbf{z}_i), \tag{2}$$

where $i$ is the index of the sample, $\mathbf{z}_i$ is the latent representation of $\mathbf{x}_i$, and $N$ is the size of the training set. By the Bayes' rule, the posterior probability can be rewritten as the following under uniform prior:

$$p(y \mid \mathbf{z}) = \frac{p(\mathbf{z} \mid y)p(y)}{p(\mathbf{z})} = \frac{p(\mathbf{z} \mid y)}{\sum_{j=1}^C p(\mathbf{z} \mid y = j)}. \tag{3}$$

Thus, one can define $S(\mathbf{z}) = \log \sum_{j=1}^C p(\mathbf{z} \mid y = j)$ as a measure of how well a given embedding matches the distribution of ID samples, which is proportional to the log-likelihood $\log p(\mathbf{z})$. Despite the connection, the challenge lies in *how to concretely define the class-conditional latent distribution $p(\mathbf{z} \mid y)$, as well as how to efficiently optimize neural networks to achieve such latent distribution while training a classifier.* In what follows, we address these two challenges.

### 3.1 Intrinsic Likelihood

To address the challenge of defining class-conditional latent distributions, we model the representations by the von Mises-Fisher (vMF) distribution in the hypersphere. A hypersphere is a topological space that is equivalent to a standard $(d-1)$-sphere, which represents the set of points in $d$-dimensional Euclidean space that are equidistant from the center (see Figure 1). The unit hypersphere is a special sphere with a unit radius and is denoted as $S^{d-1} := \{\mathbf{z} \in \mathbb{R}^d | \|\mathbf{z}\|_2 = 1\}$ in the $(d-1)$-dimensional space. The formal definition of vMF distribution is given below:

**Definition 3.1** (von Mises-Fisher Distribution (Fisher, 1953)). *For a unit vector $\mathbf{z} \in \mathbb{R}^d$ in class $c$, the conditional probability density function of the vMF distribution is defined as*

$$p(\mathbf{z} \mid y = c) = Z_d(\kappa) \exp(\kappa \boldsymbol{\mu}_c^\top \mathbf{z}), \tag{4}$$

where $\boldsymbol{\mu}_c \in \mathbb{R}^d$ denotes the mean direction of the class $c$, $\kappa \geq 0$ denotes the concentration of the distribution around $\boldsymbol{\mu}_c$, and $Z_d(\kappa)$ denotes the normalization factor.

**Benefits of the probabilistic model.** We model the latent representations as vMF distribution because it is a simple and expressive probability distribution in directional statistics. Compared with multivariate Gaussian distribution, vMF distribution avoids estimating large covariance matrices for high-dimensional data that is shown to be costly and unstable in literature (Chen et al., 2017). Meanwhile, choosing the vMF distribution allows us to have the features live on the unit hypersphere, which leads to several benefits. For example, fixed-norm vectors are known to improve training stability in modern machine learning, where dot products are ubiquitous and are beneficial for learning a good representation space. For this reason, hyperspherical representations have been popularly adopted in recent contrastive learning literature (Chen et al., 2020a; Khosla et al., 2020).

Under this probabilistic model, an embedding $\mathbf{z}$ is assigned to the class $c$ with the following probability

$$
\begin{aligned}
p(y = c \mid \mathbf{z}; \{\kappa, \boldsymbol{\mu}_j\}_{j=1}^C) &= \frac{Z_d(\kappa) \exp(\kappa \boldsymbol{\mu}_c^\top \mathbf{z})}{\sum_{j=1}^C Z_d(\kappa) \exp(\kappa \boldsymbol{\mu}_j^\top \mathbf{z})} \\
&= \frac{\exp(\boldsymbol{\mu}_c^\top \mathbf{z}/\tau)}{\sum_{j=1}^C \exp(\boldsymbol{\mu}_j^\top \mathbf{z}/\tau)},
\end{aligned}
\tag{5}
$$

where $\tau = 1/\kappa$ denotes a temperature parameter.

Now, we are ready to draw a novel connection between the hyperspherical distribution and the intrinsic likelihood, which is formalized below.

**Definition 3.2 (Intrinsic Likelihood Score).** *We define the intrinsic likelihood function over $\mathbf{z} \in \mathbb{R}^d$ in terms of the log partition function (or denominator in Eq. 5):*

$$
S(\mathbf{z}; \{\tau, \boldsymbol{\mu}_j\}_{j=1}^C) = \tau \cdot \log \sum_{j=1}^C \exp(\boldsymbol{\mu}_j^\top \mathbf{z}/\tau).
\tag{6}
$$

Theorem 3.1 shows that our scoring function in Eq. 6 can be functionally equivalent to the log-likelihood score, for OOD detection purposes.

**Theorem 3.1.** *Under uniform class prior, the intrinsic likelihood score $S(\mathbf{z})$ is a logarithmic function of the density $p(\mathbf{z})$, with a constant difference. The two measurements return the same level set for OOD detection.*

*Proof.* The likelihood $p(\mathbf{z})$ is the probability of observing a given embedding $\mathbf{z}$ under the vMF distribution. $p(\mathbf{z})$ can be calculated as a summation of class-conditional likelihood $p(\mathbf{z} \mid y = j)$, weighted by the class prior:

$$
\begin{aligned}
p(\mathbf{z}) &= \sum_{j=1}^C p(\mathbf{z} \mid y = j) p(y = j) \\
&= \frac{Z_d(\kappa)}{C} \sum_{j=1}^C \exp(\boldsymbol{\mu}_j^\top \mathbf{z}/\tau),
\end{aligned}
\tag{7}
$$

where the right-hand side is obtained by plugging in Eq. 4 and class prior $1/C$. By connecting the intrinsic likelihood $S(\mathbf{z})$ in Eq. 6 and the density function $p(\mathbf{z})$ in Eq. 7, we can show the following equivalence between the two:

$$
S(\mathbf{z}; \{\tau, \boldsymbol{\mu}_j\}_{j=1}^C) = \tau \cdot \underbrace{\log p(\mathbf{z})}_{\text{log likelihood of } \mathbf{z}} + \underbrace{\tau \cdot \log \frac{C}{Z_d(1/\tau)}}_{\text{const, independent of } \mathbf{z}}
\tag{8}
$$

Note that the second term is independent of $\mathbf{z}$, and can be treated as a constant. Thus, $S(\mathbf{z})$ is functionally equivalent to the log-likelihood score, for OOD detection.

**Intrinsic likelihood for OOD detection.** Given the above equivalence between INK score and log-likelihood, we propose the following OOD detector:

$$g_\lambda(\mathbf{x}) = \mathbb{1}\{S(\mathbf{z}) \geq \lambda\}, \tag{9}$$

where samples with higher scores are classified as ID, and vice versa. The threshold $\lambda$ is chosen based on the ID score at a certain percentile (e.g., 95%).

**Generalization to class-imbalanced datasets.** In Theorem 3.1, we adopt the commonly assumed scenario prevalent in OOD detection literature, where datasets are class-balanced with a uniform prior distribution over classes, i.e., $p(y = c) = 1/C$. However, we acknowledge that this assumption may not always hold in real-world scenarios.

To further demonstrate the generality and effectiveness of our method, we extend it to account for non-uniform class priors. We perform standard maximum likelihood estimation on the training dataset $\{(\mathbf{x}_i, y_i)\}_{i=1}^N$:

$$\text{argmax}_\theta \prod_{i=1}^N p(y = y_i \mid \mathbf{z}_i; \{\kappa, \boldsymbol{\mu}_j\}_{j=1}^C), \tag{10}$$

where $i$ is the index of the sample, $j$ is the index of the class, and $N$ is the size of the training set. The posterior probability is given by:

$$p(y = c \mid \mathbf{z}; \{\kappa, \boldsymbol{\mu}_j\}_{j=1}^C) = \frac{p(y = c) \exp(\boldsymbol{\mu}_c^\top \mathbf{z}/\tau)}{\sum_{j=1}^C p(y = j) \exp(\boldsymbol{\mu}_j^\top \mathbf{z}/\tau)}, \tag{11}$$

which accounts for the class prior probability, allowing for non-uniform class distributions. We define the generalized intrinsic likelihood as:

$$S(\mathbf{z}; \{\tau, \boldsymbol{\mu}_j\}_{j=1}^C) = \tau \cdot \log \sum_{j=1}^C p(y = j) \exp(\boldsymbol{\mu}_j^\top \mathbf{z}/\tau), \tag{12}$$

which maintains functional equivalence to the log-likelihood for OOD detection, even when the class priors are not uniform.

## 3.2 Optimizing Intrinsic Likelihood in Deep Neural Networks

So far, we have established that intrinsic likelihood derived under the vMF distribution is desirable for OOD detection. Now, we discuss how to optimize neural networks for intrinsic likelihood.

**Maximum likelihood estimation.** We aim to train the neural network, such that the hyperspherical embedding conforms to a mixture of vMF distributions defined in Eq. 5. Specifically, we consider a deep neural network $h : \mathcal{X} \mapsto \mathbb{R}^d$ which encodes an input $\mathbf{x} \in \mathcal{X}$ to a normalized embedding $\mathbf{z} := h(\mathbf{x})$. To optimize for the vMF distribution, one can directly perform standard maximum likelihood estimation on the training dataset $\{(\mathbf{x}_i, y_i)\}_{i=1}^N$:

$$\text{argmax}_\theta \prod_{i=1}^N p(y = y_i \mid \mathbf{z}_i; \{\kappa, \boldsymbol{\mu}_j\}_{j=1}^C), \tag{13}$$

where $i$ is the index of the sample, $j$ is the index of class, and $N$ is the size of the training set. In effect, this loss encourages each ID sample to have a high probability assigned to the correct class in the mixtures of the vMF distributions. While hyperspherical learning algorithms have been studied (Mettes et al., 2019; Du et al., 2022a; Ming et al., 2023), *none of the works explored its principled connection to intrinsic likelihood for test-time OOD detection, which is our distinct focus. We discuss the differences further in Section 3.3.* Moreover, we show new insights below that minimizing this loss function effectively shapes the intrinsic likelihood to aid OOD detection.

**How does the loss function shape intrinsic likelihood?** We show that the learning objective can induce a higher intrinsic likelihood for ID data. By taking negative-log likelihood, the objective function in Eq. 13 is equivalent to minimizing the following loss:

$$\mathcal{L} = \frac{1}{N}\sum_{i=1}^{N} -\log p(y = y_i \mid \mathbf{z}_i; \{\tau, \boldsymbol{\mu}_j\}_{j=1}^{C}). \tag{14}$$

We can rewrite the loss by expanding Eq. 14 :

$$
\begin{aligned}
\mathcal{L} &= \frac{1}{N}\sum_{i=1}^{N}\left(-\log\frac{\exp(\boldsymbol{\mu}_{y_i}^{\top}\mathbf{z}_i/\tau)}{\sum_{j=1}^{C}\exp(\boldsymbol{\mu}_j^{\top}\mathbf{z}_i/\tau)}\right)\\
&= \frac{1}{N}\sum_{i=1}^{N}\left(\frac{1}{\tau}s(\mathbf{z}_i, y_i) + \log\sum_{j=1}^{C}\exp(-s(\mathbf{z}_i, j)/\tau)\right),
\end{aligned}\tag{15}
$$

where $s(\mathbf{z}_i, y) = -\boldsymbol{\mu}_y^{\top}\mathbf{z}_i$. During optimization, the loss reduces $s(\mathbf{z}_i, y)$ for the correct class while increasing it for other classes. This can be justified by analyzing the gradient of the loss function with respect to the model parameters $\boldsymbol{\theta}$, for an embedding $\mathbf{z}$ and its associated class label $y$:

$$
\begin{aligned}
\frac{\partial\mathcal{L}(\mathbf{z}, y; \boldsymbol{\theta})}{\partial\boldsymbol{\theta}} &= \frac{1}{\tau}\frac{\partial s(\mathbf{z}, y)}{\partial\boldsymbol{\theta}} \tag{16}\\
&\quad -\frac{1}{\tau}\sum_{j=1}^{C}\frac{\partial s(\mathbf{z}, j)}{\partial\boldsymbol{\theta}}\cdot\frac{\exp(-s(\mathbf{z}, j)/\tau)}{\sum_{c=1}^{C}\exp(-s(\mathbf{z}, c)/\tau)}\\
&= \frac{1}{\tau}\underbrace{\frac{\partial s(\mathbf{z}, y)}{\partial\boldsymbol{\theta}}(1 - p(Y = y \mid \mathbf{z}))}_{\downarrow \text{ for } y} - \underbrace{\sum_{j\neq y}\frac{\partial s(\mathbf{z}, j)}{\partial\boldsymbol{\theta}}p(Y = j \mid z)}_{\uparrow \text{ for } j \neq y} \tag{17}
\end{aligned}
$$

The sample-wise intrinsic likelihood $S(\mathbf{z})$ of ID data is $S(\mathbf{z}) = \tau\cdot\log\sum_{c=1}^{C}\exp(-s(\mathbf{z}, c)/\tau)$, which is dominated by the term $-s(\mathbf{z}, y)$ with ground truth label. Hence, the training overall induces a higher intrinsic likelihood for ID data. By further connecting to our Theorem 3.1, this higher intrinsic likelihood can directly translate into high ($\uparrow$) $\log p(\mathbf{z})$, for training data distribution.

## 3.3 Differences w.r.t. Existing Approaches

Table 1 summarizes the key distinctions among prevalent hyperspherical methodologies, specifically in terms of their training-time loss functions and test-time scoring functions. Both SSD+ (Sehwag et al., 2021) and KNN+ (Sun et al., 2022b) employ the SupCon (Khosla et al., 2020) loss during training, which does not explicitly model the latent representations as vMF distributions. Instead of promoting instance-to-prototype similarity, SupCon promotes instance-to-instance similarity among positive pairs. SupCon's loss formulation thus does not directly correspond to the vMF distribution. Geometrically speaking, our framework directly operates on the vMF distribution, a key property that enables log-likelihood interpretation.

Our approach also differs significantly from CIDER (Ming et al., 2023) and SIREN (Du et al., 2022a) in terms of OOD scoring function, which employ either KNN distance or maximum conditional probability $\max p(y|\mathbf{z})$. In contrast, our method establishes the novel connection between the hyperspherical representations and intrinsic likelihood for OOD detection, which enjoys rigorous theoretical interpretation from a density perspective. We show empirically in Section 4 that our proposed OOD score achieves competitive performance on different OOD detection benchmarks, and is much more computationally efficient by eliminating the need for time-consuming KNN searches.

Table 1: Summary of key distinction among different hyperspherical approaches.

| | Training-time loss function | Test-time scoring function | Likelihood interpretation |
|---|---|---|---|
| SSD+ | SupCon | Mahalanobis (parametric) | ✗ |
| KNN+ | SupCon | KNN (non-parametric) | ✗ |
| CIDER | vMF | KNN (non-parametric) | ✗ |
| SIREN | vMF | KNN or $\max p(y\|\mathbf{z})$ | ✗ |
| INK (Ours) | vMF | Intrinsic likelihood (parametric) | ✓ |

## 4 Experiments

### 4.1 Setup

**Benchmarks.** Our evaluation encompasses both far-OOD and near-OOD detection performance, following the same setup as the latest OpenOOD benchmark[1] (Zhang et al., 2023a). The benchmark includes CIFAR (Krizhevsky, 2009) and ImageNet-1k (Deng et al., 2009) as ID datasets. Due to space constraints, we primarily focus on the most challenging ImageNet benchmark in the main paper, and report the full results for CIFAR datasets in the Appendix E. For the ImageNet benchmark, iNaturalist (Van Horn et al., 2018), Textures (Cimpoi et al., 2014), and OpenImage-O (Wang et al., 2022) are employed as far-OOD datasets. In addition, we also evaluate on the latest NINCO (Bitterwolf et al., 2023) as the near-OOD dataset, which circumvents the issue of contaminated classes found in other near-OOD datasets (Vaze et al., 2022). We adhere to the same data split used in OpenOOD (Zhang et al., 2023a), which involves the removal of images with semantic overlap.

**Evaluation metrics and implementation details.** We utilize the following metrics, which are widely used to assess the efficacy of OOD detection methods: (1) false positive rate (FPR@95) of OOD samples when the true positive rate of ID samples is set at 95%, (2) area under the receiver operating characteristic curve (AUROC), and (3) ID classification accuracy (ID ACC). Due to the space limit, we provide the implementation details in the Appendix B and ID accuracy in Appendix D. For INK score, we use test-time temperature $\tau_{\text{test}} = 0.05$ based on validation (see Appendix C). We further show the effect of the temperature in our ablation study (Section 4.3).

### 4.2 Main Results

**Intrinsic likelihood score establishes state-of-the-art performance.** We extensively compare the performance of our approach with state-of-the-art methods. The compared methods are categorized as either hyperspherical-based or non-hyperspherical-based. All the non-hyperspherical-based methods, including MSP (Hendrycks & Gimpel, 2017), ODIN (Liang et al., 2018), Mahalanobis (Lee et al., 2018b), Energy (Liu et al., 2020), ViM (Wang et al., 2022), ReAct (Sun et al., 2021), DICE (Sun & Li, 2022), SHE (Zhang et al., 2023b), and ASH (Djurisic et al., 2023) use the softmax cross-entropy (CE) loss for model training. Hyperspherical-based methods, on the other hand, include SSD+ (Sehwag et al., 2021), KNN+ (Sun et al., 2022b), and CIDER (Ming et al., 2023). It is worth noting that we have adopted the recommended configurations proposed by prior works. Further details of each baseline method are included in Appendix B.

As shown in Table 2, our method INK displays competitive OOD detection performance compared to existing ones on the far-OOD group of the ImageNet benchmark. Here we highlight two observations: (1) INK also outperforms the current state-of-the-art cross-entropy loss method ASH (Djurisic et al., 2023), which employs a heuristic-driven activation shaping approach and lacks a rigorous likelihood interpretation. In contrast, INK is derived from a more principled likelihood-based perspective while being empirically strong. (2) INK score outperforms the current state-of-the-art hyperspherical-based methods CIDER by 8.97% and KNN+ by 6.60% in FPR@95. Compared to CIDER, our method enjoys a significant performance boost by leveraging the likelihood interpretation within hyperspherical space, and a drastic reduction in computational cost (more in Section 4.3).

---

[1] https://github.com/Jingkang50/OpenOOD/

Table 2: **OOD detection performance for ImageNet (ID) with ResNet-50, using OpenOOD benchmark (Zhang et al., 2023a).** INK achieves competitive performance compared to state-of-the-art methods. Statistical significance is estimated on 3 random independent runs.

| Method | Far-OOD Datasets | | | | | | | | Near-OOD Dataset | |
| | iNaturalist | | Textures | | OpenImage-O | | Average | | NINCO | |
| | FPR ↓ | AUROC ↑ | FPR ↓ | AUROC ↑ | FPR ↓ | AUROC ↑ | FPR ↓ | AUROC ↑ | FPR ↓ | AUROC ↑ |
|---|---|---|---|---|---|---|---|---|---|---|
| | *Methods using cross-entropy loss* | | | | | | | | | |
| MSP | 43.34 | 88.94 | 60.88 | 80.13 | 50.13 | 86.69 | 51.45 | 85.25 | 56.87 | 79.95 |
| ODIN | 36.07 | 90.63 | 49.44 | 89.06 | 46.55 | 86.58 | 44.02 | 88.76 | 68.13 | 77.77 |
| Mahalanobis | 73.80 | 63.67 | 42.77 | 69.27 | 72.15 | 51.96 | 62.91 | 61.63 | 78.83 | 62.38 |
| Energy | 31.34 | 92.54 | 45.76 | 88.26 | 38.08 | 88.96 | 38.39 | 89.92 | 60.58 | 79.70 |
| ViM | 30.67 | 92.65 | 10.49 | 86.52 | 32.82 | 90.78 | 24.66 | 89.98 | 62.28 | 78.63 |
| ReAct | 16.73 | 96.34 | 29.63 | 91.87 | 32.57 | 93.41 | 26.31 | 93.87 | 55.91 | 81.73 |
| DICE | 33.38 | 91.17 | 44.25 | 88.24 | 47.83 | 86.77 | 41.82 | 88.72 | 66.97 | 76.01 |
| SHE | 34.03 | 89.56 | 35.30 | 90.50 | 54.98 | 83.33 | 41.44 | 87.80 | 69.75 | 76.48 |
| ASH | 14.10 | 97.06 | 15.27 | 93.26 | 29.20 | 93.29 | 19.52 | 94.54 | 52.99 | 83.44 |
| | *Methods using hyperspherical representation* | | | | | | | | | |
| SSD+ | 39.82 | 88.41 | 71.90 | 84.86 | 59.05 | 81.65 | 56.92 | 84.97 | 61.74 | 80.61 |
| KNN+ | 18.29 | 94.90 | 19.90 | 93.56 | 27.40 | 89.05 | 21.87 | 92.50 | 53.35 | 84.15 |
| CIDER | 20.79 | 94.39 | 21.86 | 92.54 | 30.06 | 89.07 | 24.24 | 92.00 | 57.42 | 82.36 |
| INK (ours) | $10.04^{\pm0.67}$ | $97.49^{\pm0.08}$ | $7.52^{\pm2.08}$ | $98.62^{\pm0.51}$ | $28.26^{\pm0.90}$ | $94.24^{\pm0.23}$ | $15.27^{\pm0.27}$ | $96.78^{\pm0.12}$ | $48.15^{\pm1.33}$ | $86.42^{\pm0.13}$ |

Table 3: **Average and standard deviation of inference time (per sample) for each ID dataset.** The INK score is computationally more efficient compared to the KNN search.

| Score | Score compute time ($\mu$s↓) | | | |
| | CIFAR-10 | CIFAR-100 | ImageNet-1k ($\alpha = 1\%$) | ImageNet-1k ($\alpha = 100\%$) |
|---|---|---|---|---|
| KNN+ or CIDER | $3.5619 \pm 0.2646$ | $5.4886 \pm 0.1854$ | $3.6157 \pm 0.1186$ | $336.1780 \pm 4.3871$ |
| INK (ours) | $\mathbf{0.0076} \pm 0.0003$ | $\mathbf{0.0157} \pm 0.0007$ | $\mathbf{0.2745} \pm 0.0145$ | $\mathbf{0.2745} \pm 0.0145$ |

**Near-OOD detection.**  We also evaluate the performance of our method in the more challenging near-OOD scenario. As shown in Table 2, INK demonstrates a competitive performance, outperforming CIDER by 9.27% in FPR@95. This improvement underscores the versatility of the INK score, proving its effectiveness in near-OOD detection scenarios where out-of-distribution samples bear a close semantic resemblance to the ID samples. Moreover, our method shows an enhanced performance over ASH in near-OOD detection, with an improvement of 4.84% in FPR@95. This advancement also signifies progress in bridging the performance gap between hyperspherical-based methods and cross-entropy loss methods.

**CIFAR benchmark.**  We evaluate the performance of our method on the CIFAR benchmark. Consistent with the results observed in the ImageNet benchmark, the INK score displays competitive performance in handling both far-OOD and near-OOD groups, compared to the current state-of-the-art methods. Due to space constraints, further details are included in Appendix E.

## 4.3   Additional Ablations and Discussions

**Computational cost.**  The current state-of-the-art hyperspherical representation methods, such as KNN+ and CIDER, rely on a non-parametric KNN score, which requires a nearest neighbor search in the embedding space. Although recent studies on approximated nearest neighbor search have shown promise (Aumüller et al., 2020), the computation required grows with the size of the embedding pool $N$. In contrast, INK only incurs complexity $O(C)$, where the number of classes $C \ll N$ is independent of ID sample size. Thus, our method can significantly reduce the computation overhead. To demonstrate this, we conduct an experiment to compare the average compute time between the KNN score and the INK score. In this experiment, we assess the performance of the model on CIFAR-10, CIFAR-100, and ImageNet-1k benchmarks using default values of $k$ in Sun et al. (2022b). In particular, we measure the time taken to calculate the score per sample, excluding the time to extract the sample embedding from the neural network. KNN-based methods utilize the Faiss library (Johnson et al., 2019), which is optimized for efficient nearest-neighbor search. The size of the embedding pool is 50,000 for both CIFAR-10 and CIFAR-100, 12K for ImageNet ($\alpha = 1\%$), and 1.2M for

ImageNet ($\alpha = 100\%$), where $\alpha$ denotes the fraction of training data used for the nearest neighbor search. Our results, shown in Table 3, demonstrate that INK is significantly faster than the KNN-based methods, making it computationally desirable for OOD detection tasks.

**Relationship with energy score.** We discuss the relationship of INK with prior work by Liu *et al.* Liu et al. (2020), which attempted to interpret the energy score derived from a multi-class classifier from a likelihood-based view. The key distinction between the two methods lies in the geometry of the embedding space, from which the score is derived. Specifically, an energy score is defined as the negative log of the denominator in the softmax function:

$$E(\mathbf{x}; f_{\boldsymbol{\theta}}) = -\tau \cdot \log \sum_{j=1}^{C} \exp(f_{\boldsymbol{\theta}}^j(\mathbf{x})/\tau),$$

where $f_{\boldsymbol{\theta}}^j(\mathbf{x})$ denotes the logit corresponding to the class $j$. Unfortunately, the energy score derived from the Euclidean space logits $f_{\boldsymbol{\theta}}(\mathbf{x})$ can be *unconstrained*, and does not necessarily reflect the log-likelihood (see Theorem A.1 and proof in Appendix). In contrast, our framework explicitly models the latent representations in the discriminative classifier

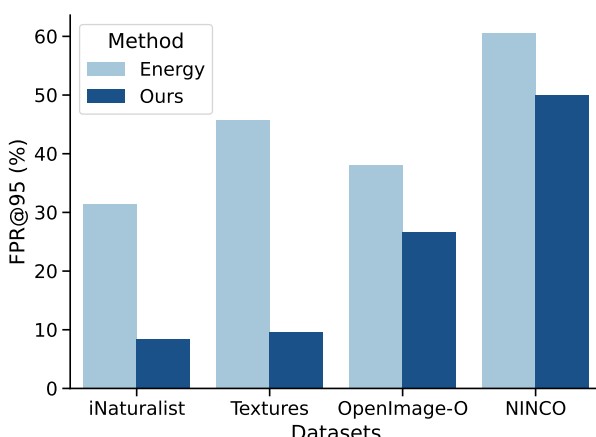

Figure 2: **Comparison of Energy vs. INK on ImageNet (ID) with ResNet-50, using OpenOOD benchmark Zhang et al. (2023a)**. INK consistently outperforms Energy on far-OOD and near-OOD datasets.

as *constrained hyperspherical space*, which enables a rigorous interpretation from a log-likelihood perspective (*c.f.* Theorem 3.1). In our formulation (Eq. 5), the classifier's logit $f_{\boldsymbol{\theta}}^j(\mathbf{x})$ is equivalent to $\boldsymbol{\mu}_j^\top \mathbf{z}$, where $\mathbf{z}$ is the penultimate layer embedding of input $\mathbf{x}$ and $\boldsymbol{\mu}_j$ can be interpreted as the weight vector connecting the embedding and output logit. Different from Liu et al. (2020), the embedding $\mathbf{z}$ and the weight connection $\boldsymbol{\mu}_j$ are explicitly constrained together to form a meaningful probabilistic distribution during our optimization.

Figure 2 illustrates a comparison of the performance on the ImageNet benchmark. Our method reduces the average FPR@95 by 23.12%, which validates that the proposed intrinsic likelihood score is significantly more effective for OOD detection. The performance improvement can also be reasoned theoretically, as we show in Theorem 3.1. Overall, our method enjoys both strong empirical performance and theoretical justification. Additionally, for a more detailed understanding, we have included a visualization of ID-OOD score distributions in Figure 4. We illustrate a comparison between the distributions of energy and INK scores for ImageNet (ID) vs. Textures (OOD). The noticeable separability of using INK score enhances the performance in OOD detection.

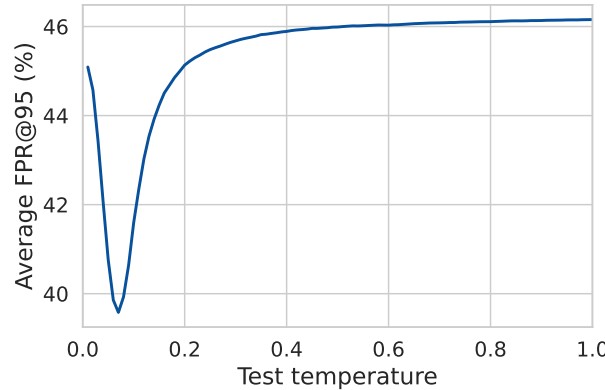

Figure 3: **Ablation on test-time temperatures.** The results are averaged across the far-OOD test sets and **3 random training runs**, based on ResNet-34 trained on CIFAR-100.

**Ablation study on test-time temperature $\tau$.**
We conduct an ablation study on the test-time temperature parameter to investigate its impact on OOD detection performance, as shown in Figure 3. The curve highlights that the optimal value of $\tau$ roughly matches the training-time temperature. This also matches our theory, where INK under the same temperature $\tau$ in testing allows recovering the log-likelihood $\log p(\mathbf{z})$ (up to some constant) learned in training time.

**Ablation study on different model architecture.** To explore the impact of model architecture on the performance, we additionally evaluate the Vision Transformer (ViT-B-16) (Dosovitskiy et al., 2021). As

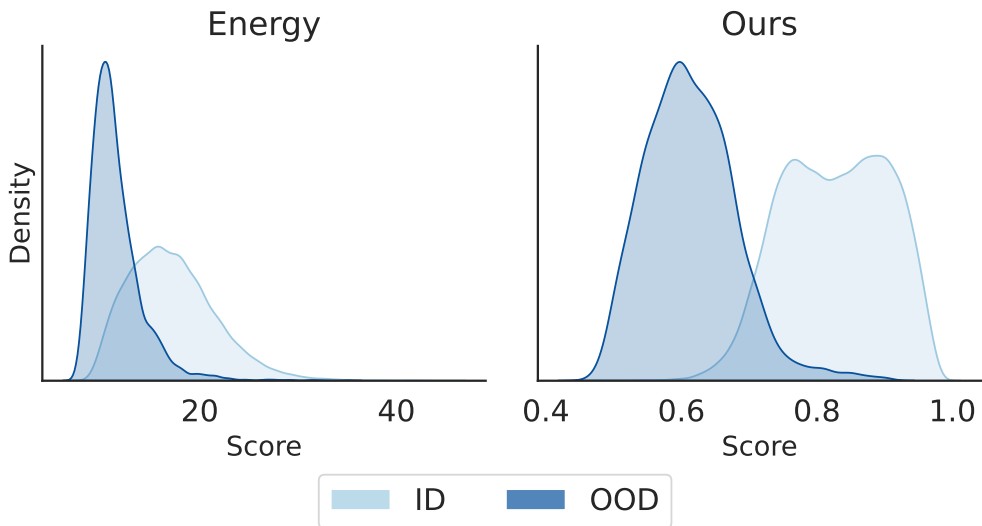

Figure 4: **Density plots illustrating the distribution of energy score (left) and INK (right).** Lighter and darker blue shades represent the distributions of scores for the ID and OOD datasets, respectively.

shown in Table 7 in Appendix, our INK score maintains competitive performance with the ViT model. This underscores the adaptability of our method across various model architectures. Further details are included in Appendix F.

**Ablation study on class-imbalance scenarios.**   We further verify this class-imbalanced setting empirically by eliminating half of the training examples from the first 5 classes in CIFAR-10 and the first 10 classes in CIFAR-100. Table 4 compares the FPR@95 between ASH and the generalized intrinsic likelihood, which shows that our method significantly outperforms the state-of-the-art in both far-OOD and near-OOD detection tasks.

Table 4: **Comparison of OOD Performance with Imbalanced Class Prior.** Generalized INK achieves competitive performance compared to state-of-the-art method ASH across different datasets.

<table>
<tr><td colspan="3">**CIFAR-10 Imbalanced (ResNet-18)**</td><td colspan="3">**CIFAR-100 Imbalanced (ResNet-34)**</td></tr>
<tr><td>**Dataset**</td><td>**ASH**</td><td>**Ours**</td><td>**Dataset**</td><td>**ASH**</td><td>**Ours**</td></tr>
<tr><td>**Far-OOD**</td><td></td><td></td><td>**Far-OOD**</td><td></td><td></td></tr>
<tr><td>MNIST</td><td>25.60</td><td>23.97</td><td>MNIST</td><td>48.29</td><td>37.72</td></tr>
<tr><td>SVHN</td><td>52.73</td><td>9.12</td><td>SVHN</td><td>33.19</td><td>10.47</td></tr>
<tr><td>Texture</td><td>42.00</td><td>18.69</td><td>Texture</td><td>55.31</td><td>34.29</td></tr>
<tr><td>Places365</td><td>40.28</td><td>26.12</td><td>Places365</td><td>67.81</td><td>62.47</td></tr>
<tr><td>*Far-OOD Average*</td><td>40.15</td><td>**19.48**</td><td>*Far-OOD Average*</td><td>51.15</td><td>**36.24**</td></tr>
<tr><td>**Near-OOD**</td><td></td><td></td><td>**Near-OOD**</td><td></td><td></td></tr>
<tr><td>CIFAR-100</td><td>53.44</td><td>40.33</td><td>CIFAR-10</td><td>74.39</td><td>76.36</td></tr>
<tr><td>TIN</td><td>48.38</td><td>28.47</td><td>TIN</td><td>65.74</td><td>53.60</td></tr>
<tr><td>*Near-OOD Average*</td><td>50.91</td><td>**34.40**</td><td>*Near-OOD Average*</td><td>70.07</td><td>**64.98**</td></tr>
</table>

# 5   Related Works

**OOD detection for classification models.**   OOD detection is an essential component for reliable machine learning systems that operate in the open world. Recent studies in OOD detection focus on post-hoc strategies

and train-time regularization (Yang et al., 2021b). In particular, post-hoc strategies focus on deriving test-time OOD scoring functions, including confidence-based scores (Hendrycks & Gimpel, 2017; Liang et al., 2018), distance-based score (Lee et al., 2018b; Tack et al., 2020; Sehwag et al., 2021; Sastry & Oore, 2020; Ren et al., 2021; Sun et al., 2022b; Du et al., 2022a; Ming et al., 2022; Ren et al., 2023), energy-based score (Liu et al., 2020; Wang et al., 2021; Morteza & Li, 2022; Zhang et al., 2023b; Jiang et al., 2023; Lafon et al., 2023; Peng et al., 2024), gradient-based score (Huang et al., 2021; Behpour et al., 2023; Chen et al., 2023; Liu et al., 2023a), and multimodal score (Ming et al., 2022; Miyai et al., 2023). Recent high-performing methods focus on suppressing activations of pre-trained models (Sun et al., 2021; Sun & Li, 2022; Djurisic et al., 2023; Xu et al., 2023b). On the other hand, train-time regularization approaches (Bevandić et al., 2018; Malinin & Gales, 2018; Lee et al., 2018a; Geifman & El-Yaniv, 2019; Hendrycks et al., 2019; Hein et al., 2019; Meinke & Hein, 2020; Van Amersfoort et al., 2020; Yang et al., 2021a; Du et al., 2022b; Wei et al., 2022; Katz-Samuels et al., 2022; Du et al., 2022a; Huang et al., 2022; Colombo et al., 2022; Yu et al., 2023; Tao et al., 2023; Ming et al., 2023; Wang et al., 2023; Xu et al., 2023a) design an algorithm to encourage the model to produce desired properties. Different from prior works, we regularize a model to produce hyperspherical representations aligned with the vMF distribution and propose intrinsic likelihood as a test-time scoring function, which has not been explored in the past. Moreover, *our method offers theoretical justification based on the likelihood view, which is lacking in previous approaches.*

**Likelihood-based OOD detection.** Deep generative models have been explored for unsupervised OOD detection (Nalisnick et al., 2019; Ren et al., 2019; Abati et al., 2019; Choi et al., 2019; Serrà et al., 2020; Xiao et al., 2020; Wang et al., 2020; Schirrmeister et al., 2020; Sastry & Oore, 2020; Kirichenko et al., 2020; Li et al., 2022; Cai & Li, 2023; Liu et al., 2023b). While these approaches directly estimate the likelihood of ID data, the unsupervised nature *prohibits classification ability.* Moreover, issues have been highlighted that deep generative models can frequently assign spuriously higher likelihoods to OOD than ID samples (Ren et al., 2019), and the performance can often lag behind the discriminative approaches (Yang et al., 2021b). In contrast, our method circumvents this issue by virtue of its supervised learning objective, leveraging ground truth semantic labels for each ID data point. Consequently, the latent embeddings are shaped and guided by image semantics without necessarily overfitting on background elements.

**Hyperspherical representation.** Hyperspherical representation under vMF distribution has been extensively used in various machine learning applications, such as supervised classification (Kobayashi, 2021; Scott et al., 2021; Govindarajan et al., 2023), face verification (Hasnat et al., 2017; Conti et al., 2022), generative modeling (Davidson et al., 2018), segmentation (Hwang et al., 2019; Sun et al., 2022a), and clustering (Gopal & Yang, 2014). In addition, some researchers have utilized the vMF distribution for anomaly detection by employing generative models and using it as the prior for zero-shot learning (Chen et al., 2020b) and document analysis (Zhuang et al., 2017). Recent studies devise vMF-based learning for OOD detection, which offers a direct and clear geometrical interpretation of hyperspherical embeddings. Specifically, Du et al. (2022a) explores modeling representations into vMF distribution for object-level OOD detection, while CIDER (Ming et al., 2023) shows that additional regularization for large angular distances among different class prototypes is crucial for achieving strong OOD detection performance. Our work adopts a vMF-based learning approach for OOD detection and builds upon these recent studies by regularizing a model to produce representations that align with the vMF distribution and proposing a new intrinsic likelihood scoring function for test-time OOD detection.

## 6 Conclusion

In this paper, we introduce intrinsic likelihood for detecting out-of-distribution samples. Our approach is based on density estimation in the hyperspherical space, which enjoys a sound interpretation from a log-likelihood perspective. Through extensive experiments on common OOD detection benchmarks, we demonstrate the state-of-the-art performance of our approach for both far-OOD and near-OOD detection. Moreover, our method outperforms KNN-based approaches in terms of computational efficiency. Overall, our proposed intrinsic likelihood score provides a promising solution for both effective and efficient OOD detection.

## 7 Broader Impacts

OOD detection can be used to improve the reliability of machine learning models, which can have a significant positive impact on society. In situations where saving features is a concern, the use of the KNN score may not be feasible as it requires access to a certain amount of labeled data. In contrast, our INK score does not require access to a pool of ID data. This makes it a valuable tool in various applications and industries where sensitive information needs to be protected. It is important to note that the usefulness and effectiveness of OOD detection may vary across different applications and domains, and careful evaluation and validation are necessary before its deployment.

## Acknowledgment

We gratefully acknowledge the support from the AFOSR Young Investigator Program under award number FA9550-23-1-0184, National Science Foundation (NSF) Award No. IIS-2237037 & IIS-2331669, Office of Naval Research under grant number N00014-23-1-2643, Philanthropic Fund from SFF, and faculty research awards/gifts from Google and Meta.

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

## A    Derivation

**Theorem A.1** (**Misalignment between energy score and likelihood**). *The energy score defined in Equation 4.3 does not necessarily reflect the negative log probability:*

$$E(\mathbf{x}; f_{\boldsymbol{\theta}}) \not\propto -\log p(\mathbf{x}).$$

To prove this, one can rewrite the posterior probability $p_{\boldsymbol{\theta}}(y = c|\mathbf{x}) = p_{\boldsymbol{\theta}}(\mathbf{x}, y = c)/p_{\boldsymbol{\theta}}(\mathbf{x})$. Note that one can always rescale the logits $f_{\boldsymbol{\theta}}(\mathbf{x})$ by a function $\log \phi(\mathbf{x})$, without changing the posterior probability: $p_{\boldsymbol{\theta}}(y = c|\mathbf{x}) = p_{\boldsymbol{\theta}}(\mathbf{x}, y = c)\phi(\mathbf{x})/p_{\boldsymbol{\theta}}(\mathbf{x})\phi(\mathbf{x})$. In this case, the negative log of the denominator in softmax (i.e., the energy score) is $-\log[p_{\boldsymbol{\theta}}(\mathbf{x})\phi(\mathbf{x})]$, which is no longer truthfully reflecting the $-\log p(\mathbf{x})$.

## B    Experimental Details

**Software and hardware.** We conduct our experiments on NVIDIA RTX A6000 GPUs (48GB VRAM). We use Ubuntu 22.04.2 LTS as the operating system and install the NVIDIA CUDA Toolkit version 11.6 and cuDNN 8.9. All experiments are implemented in Python 3.8 using the PyTorch 1.8.1 framework.

**Training cross-entropy models.** For methods using cross-entropy loss, such as MSP (Hendrycks & Gimpel, 2017), ODIN (Liang et al., 2018), Mahalanobis (Lee et al., 2018b), Energy (Liu et al., 2020), ViM (Wang et al., 2022), ReAct (Sun et al., 2021), DICE (Sun & Li, 2022), SHE (Zhang et al., 2023b), and ASH (Djurisic et al., 2023), the models are trained using stochastic gradient descent with momentum 0.9 and weight decay $10^{-4}$. The initial learning rate is 0.1 and decays by a factor of 10 at epochs 100, 150, and 180. We train the models for 200 epochs on CIFAR. For the ImageNet benchmark, we adopt Torchvision's pre-trained ResNet-50 model with ImageNet-1k weights.

**Training contrastive models.** For methods using SupCon loss (Khosla et al., 2020), such as KNN+ (Khosla et al., 2020) and SSD+ (Sehwag et al., 2021), we adopt the same training scheme as in (Ming et al., 2023). For CIFAR, we use stochastic gradient descent with a momentum of 0.9 and a weight decay of $10^{-4}$. The initial learning rate to 0.5 and follows a cosine annealing schedule. We train the models for 500 epochs. The training-time temperature $\tau$ is set to be 0.1. For the ImageNet model, we use the checkpoint provided in Sun et al. (2022b).

For methods using vMF loss, we train on CIFAR using stochastic gradient descent with momentum 0.9 and weight decay $10^{-4}$. The initial learning rate is 0.5 and follows a cosine annealing schedule. We use a batch size of 512 and train the model for 500 epochs. The training-time temperature $\tau$ is set to be 0.1. We adopt the exponential-moving-average (EMA) for the prototype update Li et al. (2020), with a momentum of 0.5 for CIFAR-100. For ImageNet-1k, we fine-tune a pre-trained ResNet-50 model in Khosla et al. (2020) for 100 epochs, with an initial learning rate of 0.01 and cosine annealing schedule.

**Evaluation configurations.**

We outline the configurations for methods that require appropriate hyperparameter selection, which has already been extensively studied in the literature.

- MSP uses the maximum softmax probability obtained from the logits. The method does not require any specific configuration.

- ODIN uses temperature scaling to calibrate the softmax score. We set the temperature $T$ to 1,000.

- Mahalanobis utilizes class conditional Gaussian distributions based on the low- and upper-level features obtained from a model. These distributions are then used to calculate the Mahalanobis distance.

- Energy derives the energy score, which includes the temperature parameter. We set the temperature to the default value of $T = 1$.

- ViM generates an additional logit from the residual of the feature against the principal space. We set the dimension of principal space to $D = 256$ for ResNet-18 and ResNet-34 and $D = 1024$ for ResNet-50.

- ReAct improves the energy score by rectifying activations at an upper limit, which is set based on the $p$-th percentile of the activations estimated on the in-distribution (ID) data. We set $p$ to the default value of $p = 90$.

- DICE utilizes logit sparsification to enhance the vanilla energy score, which is set based on the $p$-th percentile of the unit contributions estimated on the ID data. We set the sparsity parameter $p = 0.7$.

- SHE stores the mean direction of the penultimate layer features from correctly classified training samples. During inference, the Hopfield energy score is calculated as the dot product between the sample embedding and the class mean of the predicted class.

- KNN+ and CIDER utilize the KNN score, which requires selecting a number of nearest neighbors $k$. Following the settings in Ming et al. (2023), we set $k$ to 300 and 1,000 for CIFAR-100 and the ImageNet benchmark, respectively.

- ASH proposes three heuristic-based approaches for activation shaping: Pruning, Binarizing, and Scaling. In line with the OpenOOD framework, we adopt the Binarizing method (ASH-B), specifically with a parameter setting of $p = 90$.

## C   Validation Method for Selecting Test-time Temperature

To select the test-time temperature for our proposed INK score, we follow the validation method outlined in Hendrycks et al. (2019). We generate a validation distribution by corrupting in-distribution data with speckle noise, creating speckle-noised anomalies that simulate out-of-distribution data. After that, we compute the performance of INK at different test-time temperatures on the validation set and select the one that achieves the highest AUROC.

## D   ID Classification Accuracy

Table 5 presents the in-distribution classification accuracy for each training dataset. We evaluate the classification accuracy by performing linear probing on normalized features, following the approach in Khosla et al. (2020). Using vMF loss shows competitive ID classification accuracy compared to standard cross-entropy loss, indicating it does not compromise the model's capability to distinguish samples between in-distribution classes.

Table 5: ID classification accuracy on CIFAR-100 and ImageNet (%).

| Method | ID classification accuracy ↑ | |
| --- | --- | --- |
| | CIFAR-100 | ImageNet-1k |
| Cross-entropy | 75.03 | 76.15 |
| vMF | 74.28 | 76.55 |

## E   Results on CIFAR Benchmark

In this section, we present additional results and analysis on the CIFAR-100 benchmarks. Specifically, we utilize CIFAR-10 and TIN as the near-OOD datasets, while MNIST (Deng, 2012), SVHN (Netzer et al., 2011), Textures (Cimpoi et al., 2014), and Places (Zhou et al., 2016) serve as our far-OOD datasets. As shown in Table 6, INK displays competitive performance compared to existing state-of-the-art methods. INK achieves average FPR@95 values of 44.51% and 61.83% for the far-OOD and near-OOD groups. As shown in Table 6, INK shows strong performance on the far-OOD benchmark (average FPR 44.51%), outperforming

state-of-the-art methods for large distribution shifts. Performance on near-OOD, however, is comparable across methods, likely due to CIFAR's lower resolution compared to ImageNet-1K. Compared to KNN+ and CIDER, our method achieves competitive performance **while reducing the inference time by more than 300 times** (Section 4.3).

Table 6: **OOD detection performance for CIFAR-100 (ID) with ResNet-34.** INK achieves competitive performance compared to state-of-the-art methods.

| Method | MNIST | | SVHN | | Far-OOD Datasets Textures | | Places365 | | Average | | CIFAR-10 | | Near-OOD Dataset TIN | | Average | |
|---|---|---|---|---|---|---|---|---|---|---|---|---|---|---|---|---|
| | FPR ↓ | AUROC ↑ | FPR ↓ | AUROC ↑ | FPR ↓ | AUROC ↑ | FPR ↓ | AUROC ↑ | FPR ↓ | AUROC ↑ | FPR ↓ | AUROC ↑ | FPR ↓ | AUROC ↑ | FPR ↓ | AUROC ↑ |
| | | | | | | | | Methods using cross-entropy loss | | | | | | | | |
| MSP | 65.59 | 72.26 | 60.33 | 74.38 | 71.97 | 72.12 | 58.68 | 77.07 | 64.14 | 73.96 | 62.82 | 75.89 | 54.87 | 79.23 | 58.84 | 77.56 |
| ODIN | 49.66 | 82.27 | 61.34 | 75.56 | 70.58 | 74.30 | 59.71 | 78.67 | 60.32 | 77.70 | 62.74 | 77.04 | 56.67 | 80.63 | 59.70 | 78.84 |
| Mahalanobis | 81.44 | 57.61 | 54.07 | 81.79 | 52.73 | 87.46 | 83.37 | 60.84 | 67.90 | 71.93 | 88.04 | 54.39 | 79.83 | 63.20 | 83.94 | 58.80 |
| Energy | 57.12 | 77.17 | 49.97 | 80.90 | 70.74 | 72.40 | 58.66 | 78.34 | 59.12 | 77.20 | 64.14 | 77.51 | 54.48 | 81.26 | 59.31 | 79.39 |
| ViM | 59.28 | 71.20 | 38.34 | 88.67 | 46.42 | 88.20 | 63.41 | 75.73 | 51.86 | 80.95 | 72.01 | 73.31 | 55.71 | 80.11 | 63.86 | 76.71 |
| ReAct | 60.37 | 76.35 | 49.61 | 81.71 | 59.58 | 76.28 | 58.67 | 78.58 | 57.06 | 78.23 | 63.71 | 77.24 | 54.71 | 81.26 | 59.21 | 79.25 |
| DICE | 51.62 | 81.48 | 36.59 | 88.63 | 69.66 | 73.52 | 60.27 | 77.46 | 54.54 | 80.27 | 60.53 | 77.40 | 58.01 | 79.36 | 59.27 | 78.38 |
| SHE | 52.42 | 79.46 | 58.78 | 77.27 | 78.68 | 66.80 | 68.64 | 70.36 | 64.63 | 73.47 | 65.31 | 75.10 | 60.60 | 76.17 | 62.95 | 75.63 |
| ASH | 55.88 | 78.93 | 41.59 | 87.20 | 55.22 | 80.91 | 63.80 | 75.96 | 54.12 | 80.75 | 64.62 | 75.68 | 59.27 | 78.94 | 61.95 | 77.31 |
| | | | | | | | | Methods using hyperspherical representation | | | | | | | | |
| KNN+ | 62.34 | 75.27 | 27.31 | 92.29 | 45.41 | 85.70 | 52.60 | 81.84 | 46.91 | 83.78 | 69.22 | 76.05 | 49.38 | 83.36 | 59.30 | 79.70 |
| SSD+ | 58.57 | 75.15 | 44.86 | 80.69 | 68.37 | 76.45 | 57.06 | 77.42 | 57.22 | 77.43 | 72.27 | 72.89 | 56.41 | 79.41 | 64.34 | 76.15 |
| CIDER | 63.79 | 70.24 | 23.82 | 95.95 | 46.08 | 86.73 | 59.64 | 78.15 | 48.33 | 82.77 | 74.59 | 73.33 | 51.89 | 83.08 | 63.24 | 78.20 |
| Ours | 58.60 | 74.30 | 21.41 | 96.39 | 39.80 | 90.75 | 58.24 | 79.77 | 44.51 | 85.30 | 72.77 | 74.35 | 50.90 | 84.19 | 61.83 | 79.27 |

# F  Results on Vision Transformer architecture

This section details the performance evaluation of different methods for out-of-distribution (OOD) detection using the Vision Transformer (ViT) architecture, focusing on the ImageNet-1k dataset. Our approach involves fine-tuning a pre-trained ViT-B-16 model over 100 epochs. The initial learning rate is set at 0.01, and we employ a cosine annealing schedule for optimization. The performance of various methods is compared in Table 7. INK demonstrates competitive performance in both far-OOD and near-OOD groups, with an FPR@95 rate of 38.64% and 48.80%. This result highlights the robustness and versatility of our proposed method when applied to diverse architectural models. As a note, while INK achieves competitive results, it does not consistently outperform all methods.

# G  Visualization Analysis for Large-scale Dataset

Figure 5 presents the UMAP visualization of the learned embeddings derived from a subset of ImageNet-1k classes and larger-scale out-of-distribution (OOD) datasets. These visualizations indeed reveal compact representations, where each sample appears to be effectively drawn in towards its corresponding class prototype. A notable separation between in-distribution (ID) and OOD classes is observed.

Table 7: **OOD detection performance for ImageNet (ID) with ViT-B-16.** INK achieves competitive performance compared to state-of-the-art methods.

| Method | iNaturalist | | Textures | | Far-OOD Datasets OpenImage-O | | Average | | Near-OOD Dataset NINCO | |
|---|---|---|---|---|---|---|---|---|---|---|
| | FPR ↓ | AUROC ↑ | FPR ↓ | AUROC ↑ | FPR ↓ | AUROC ↑ | FPR ↓ | AUROC ↑ | FPR ↓ | AUROC ↑ |
| | | | | Methods using cross-entropy loss | | | | | | |
| MSP | 42.42 | 88.19 | 56.44 | 85.06 | 56.11 | 84.87 | 51.99 | 86.04 | 77.35 | 78.11 |
| ODIN | 81.14 | 79.55 | 85.62 | 77.16 | 91.09 | 71.46 | 85.95 | 76.72 | 92.62 | 65.14 |
| Mahalanobis | 20.66 | 96.01 | 38.90 | 89.41 | 30.35 | 92.38 | 29.64 | 92.60 | 48.76 | 86.52 |
| Energy | 83.58 | 79.30 | 83.65 | 81.17 | 88.79 | 76.48 | 85.34 | 78.65 | 94.16 | 66.02 |
| ViM | 17.59 | 95.72 | 40.41 | 90.61 | 29.59 | 92.18 | 29.20 | 92.50 | 57.45 | 84.64 |
| ReAct | 48.22 | 86.11 | 55.87 | 86.66 | 57.68 | 84.29 | 53.26 | 85.69 | 78.50 | 75.43 |
| DICE | 47.92 | 82.51 | 54.79 | 82.21 | 52.57 | 82.23 | 51.76 | 82.32 | 81.09 | 71.67 |
| SHE | 22.17 | 93.57 | 25.65 | 92.65 | 33.59 | 91.04 | 27.14 | 92.75 | 56.01 | 84.18 |
| ASH | 97.02 | 50.63 | 98.49 | 48.53 | 94.80 | 55.52 | 96.10 | 51.56 | 95.40 | 52.52 |
| | | | | Methods using hyperspherical representation | | | | | | |
| CIDER | 26.40 | 93.27 | 48.23 | 85.94 | 34.84 | 89.87 | 36.49 | 89.69 | 47.02 | 85.22 |
| INK (ours) | 28.45 | 92.90 | 50.57 | 85.16 | 36.80 | 89.28 | 38.61 | 89.11 | 48.86 | 84.55 |

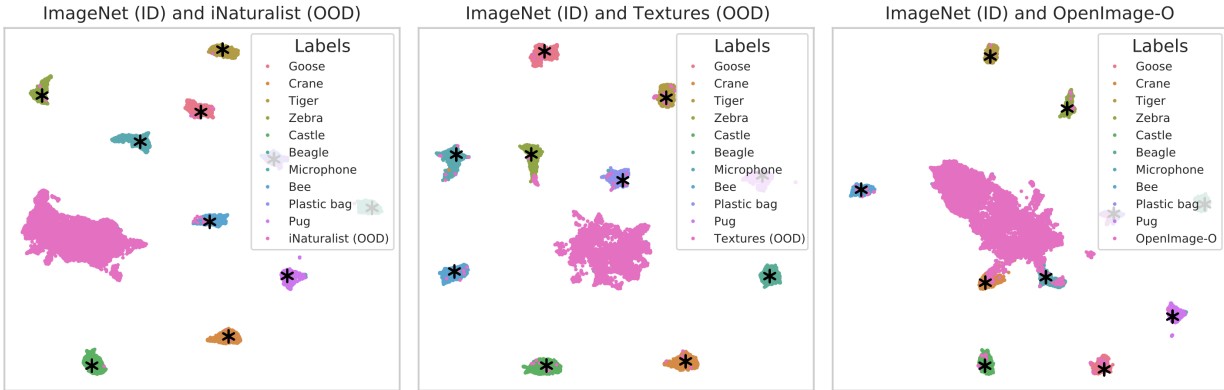

Figure 5: **UMAP visualization of a subset of ImageNet classes and OOD datasets.** The class prototypes are designated by a star symbol *, while the OOD embeddings are distinguished by pink color.

## H    Limitation

A key strength of our method is its grounded interpretation from a likelihood perspective. This approach, however, necessitates training the model under the vMF distribution. While our method achieves accuracy comparable to state-of-the-art models in certain scenarios, it's important to note that it is less common than models trained using traditional cross-entropy loss. This may limit its immediate applicability in environments where standard cross-entropy loss models are the norm. However, we believe that the trustworthiness of the models is worth the effort of training them in a certain way.

