# OpenReview forum: "Your Classifier Can Be Secretly a Likelihood-Based OOD Detector"
_TMLR — Accepted by TMLR_

### Review · Reviewer_nD6V · 2024-09-14

**Summary Of Contributions:**

The authors propose a method, **INK**, to check whether a given sample at inference time is out of distribution (OOD). By leveraging the way in which classifiers are implicitly trained to maximize the likelihood of embeddings matching in distribution (ID) data, the authors create a new, efficiently computable score for OOD detection. They evaluate **INK** on several benchmarks and against existing OOD detection methods.

**Audience:**

Yes

**Claims And Evidence:**

Yes

**Requested Changes:**

1. Explicitly describe how the attack is implemented. For example:

    * Are the models in the evaluation trained in a standard way?

    * In the implementation of **INK** is $\mu_c$ computed by taking the empirical mean of the embedding pool?


2. Describe the practical limitations of working with the assumption of embeddings being distributed over the unit hypersphere.

3. If possible, explore whether the method can be (at least heuristically) adapted to standard classification or contrastive learning settings.

**Strengths And Weaknesses:**

**Strengths**

1. The theoretical analysis uses a clever technique to do OOD detection that leverages the way in which we already train ML models.
2. Evaluation is extensive and clearly shows via many comparisons that **INK** beats previous work in both performance and efficiency at a fixed true positive rate.
3. The authors thoroughly explore other features of **INK** such as its relationship with existing work, its computational cost, sensitivity to hyperparameters, etc.

**Weaknesses**

1. Evaluation methodology for the proposed method is somewhat unclear. Although the authors *do* describe how to optimize neural networks under their framework in Section 3.2, the evaluation that follows doesn’t make it clear how this is done, or whether this is comparable to standard practices.
2. In the setting where one wants to detect OOD examples, false negatives can be costly. Given that the method is better than prior work, analysis at lower FNR (higher TPR) would be interesting (e.g. 0.1% FNR). How does the FPR change as the fixed TPR increases?
3. **Minor Comment**: The theoretical results depend on the assumption that the embeddings lie on the unit hypersphere, which is an assumption used in prior work. This enables the authors to find a nice result, but has practical constraints on the implementation of **INK**.

---

> ### Author Response · Authors · 2024-11-12
> **Response to Reviewer nD6V**
>
> Thank you for your thorough and constructive feedback on our paper. We appreciate the time and effort you put into reviewing our work. We are pleased that you found our theoretical analysis clever and our evaluation extensive. We address each of your questions below in detail:
>
>
> > **1. Evaluation details**
>
> We appreciate your concern about the clarity of our evaluation methodology. To address this, our evaluation methodology follows standard practices in OOD detection literature, as shown in the OpenOOD benchmark commonly adopted in the field [1]. We use common benchmarks (ImageNet-1k, CIFAR), metrics (FPR@95, AUROC), and architectures (ResNet-50, ResNet-34, ViT) as described in Section 4.1. **The implementation and evaluation details for all methods are described in Appendix B**.
>
> To clarify how we compute our evaluation metrics:
>
> - **FPR@95** (false positive rate at 95% true positive rate): We calculate this by first computing the OOD score for all ID and OOD samples. We then determine the score threshold that achieves a 95% true positive rate on the ID samples. Using this threshold, we compute the false positive rate on the OOD samples.
> - **AUROC** (area under the receiver operating characteristic curve): We compute this by varying the decision threshold across the full range of OOD scores and calculating the true positive rate and the false positive rate at each threshold. The AUROC is then calculated as the area under this curve.
>
> To ensure consistency and reproducibility, we use the implementation provided by the OpenOOD benchmark [1] for both the calculation of FPR95 and AUROC. This aligns our evaluation process with standard practices in OOD detection literature and allows for fair comparison across different methods.
>
> [1] Zhang et al., OpenOOD v1.5: Enhanced Benchmark for Out-of-Distribution Detection. 2023
>
> > **2. Implementation details for $\mu_c$**
>
> The implementation details for the prototype update were discussed in **Appendix B**. We adopt the exponential-moving-average (EMA) for the prototype update, following prior literature [2]. Specifically, during training, for each batch, we update $\mu_c$ as a running average of the normalized embeddings that belong to class $c$:
> $$
> \mu_c \leftarrow \alpha \mu_c + (1-\alpha) z,
> $$
> where $\alpha$ is the momentum parameter that controls the rate of update. This averaging strategy ensures that $\mu_c$ gradually aligns with the direction of the embeddings for class $c$, maintaining stability and capturing changes over time without being overly influenced by any single batch. After each update, we normalize $\mu_c$ to ensure it remains a unit vector $\|\mu_c \|_2 = 1$. This normalization step ensures that each class's mean direction vector lies on the unit hypersphere.
>
> For the INK score, the class prototypes $\mu_c$ during inference time are computed as the empirical mean of the embedding pool for each class. This can be computed once and used for all testing samples.
>
> [2] Junnan Li, Caiming Xiong, and Steven Hoi. Mopro: Webly supervised learning with momentum prototypes. In International Conference on Learning Representations, 2020.
>
> > **3. Analysis at lower FNR**
>
> We appreciate this insightful suggestion. While the AUROC metric actually captures performance across all possible thresholds, we agree that examining performance at very low false negative rates is important for many real-world applications.
>
> We have conducted additional experiments at 1% FNR (FPR@99) and 0.1% FNR (FPR@99.9). The table below provides a comparison with the most competitive baseline CIDER [3] on the ImageNet benchmark with ResNet-50 architecture.
>
> |             | **iNaturalist** | **Textures** | **OpenImage-O** | **Average Far-OOD** | **NINCO**  |
> |:----------- |:--------------- |:------------ |:--------------- |:------------------- |:---------- |
> | **Metrics** | **FPR@99**      | **FPR@99**   | **FPR@99**      | **FPR@99**          | **FPR@99** |
> | CIDER       | 48.49           | 59.26        | 56.98           | 54.91               | 76.66      |
> | INK         | 33.11           | 45.81        | 47.67           | 42.20               | 72.06      |
>
> |             | **iNaturalist** | **Textures** | **OpenImage-O** | **Average Far-OOD** | **NINCO**    |
> |:----------- |:--------------- |:------------ |:--------------- |:------------------- |:------------ |
> | **Metrics** | **FPR@99.9**    | **FPR@99.9** | **FPR@99.9**    | **FPR@99.9**        | **FPR@99.9** |
> | CIDER       | 80.05           | 83.48        | 85.58           | 83.04               | 90.12        |
> | INK         | 74.75           | 76.98        | 81.36           | 77.70               | 90.96        |
>
> These results demonstrate that INK maintains a competitive performance with its KNN score counterpart (CIDER) and an advantage on the near-OOD benchmark even at very low false negative rates.
>
>
> [3] Ming et al., How to Exploit Hyperspherical Embeddings for Out-of-Distribution Detection? ICLR 2023.

---

> > ### Author Response · Authors · 2024-11-12
> > **Response to Reviewer nD6V (cont.)**
> >
> > > **4. Hyperspherical embeddings in practice**
> >
> > The assumption of embeddings distributed over the unit hypersphere is, in practice, straightforward and highly feasible. **This constraint is easily enforced by applying an $L_2$ normalization to the embeddings, which is computationally inexpensive and can be implemented in a single line of code in most machine learning frameworks**.
> >
> > Hyperspherical embeddings have shown success in various machine learning tasks, including face verification and supervised classification (see Section 6). In particular, fixed-norm vectors are known to improve training stability in modern machine learning where dot products are ubiquitous [4], which is beneficial for learning a good representation space. For this reason, hyperspherical representations have been quite popularly adopted in recent contrastive learning literature [5,6]. Our novel contribution is to draw the bridge on how hyperspherical learning can enable principled likelihood-based OOD detection. Overall our method design thus brings (i) theoretical rigor, (ii) practical feasibility, and (iii) empirical effectiveness. We have clarified this further in our manuscript.
> >
> > [4] Wang & Isola, Understanding Contrastive Representation Learning through Alignment and Uniformity on the Hypersphere, ICML 2020.
> >
> > [5] Chen et al., A Simple Framework for Contrastive Learning of Visual Representations, ICML 2020
> >
> > [6] Khosla et al., Supervised Contrastive Learning, NeurIPS 2020
> >
> > > **5. Adaptation to standard settings**
> >
> > We appreciate this interesting suggestion. While our method is in principle designed for models trained with vMF loss, we have explored potential adaptations to more standard settings:
> >
> > - For models trained with cross-entropy loss, one could apply L2 normalization to the penultimate layer features at test time to project them onto a unit hypersphere. The INK score could then be computed using these normalized features.
> > - For contrastive learning models (SupCon), which already operate on an embeddings space, our INK score could be applied using the normalized learned embeddings and class prototypes estimated from the training data.
> >
> > The table below provides a comprehensive comparison of the ImageNet benchmark with ResNet-50 architecture.
> >
> > |                      | **iNaturalist** |           | **Textures** |           | **OpenImage-O** |           | **Average Far-OOD** |           | **NINCO**  |           |
> > |:-------------------- |:--------------- |:--------- |:------------ |:--------- |:--------------- |:--------- |:------------------- |:--------- |:---------- |:--------- |
> > | **Metrics**          | **FPR@95**      | **AUROC** | **FPR@95**   | **AUROC** | **FPR@95**      | **AUROC** | **FPR@95**          | **AUROC** | **FPR@95** | **AUROC** |
> > | Energy (CE)          | 31.34           | 92.54     | 45.76        | 88.26     | 38.08           | 88.96     | 38.39               | 89.92     | 60.58      | 79.70     |
> > | INK (CE variant)     | 36.00           | 92.47     | 20.12        | 96.27     | 79.78           | 79.61     | 45.30               | 89.44     | 87.52      | 66.99     |
> > | INK (SupCon variant) | 8.98            | 97.97     | 17.18        | 97.21     | 23.29           | 95.08     | 16.48               | 96.75     | 43.56      | 86.96     |
> > | INK                  | 10.04           | 97.49     | 7.52         | 98.62     | 28.26           | 94.24     | 15.27               | 96.78     | 48.15      | 86.42     |

---

> > > ### Comment · Reviewer_nD6V · 2024-11-27
> > > **Follow Up**
> > >
> > > Thank you for taking the time to address all of my comments. The revisions have substantially improved the paper.

---

> > > > ### Author Response · Authors · 2024-11-27
> > > >
> > > > Thank you for taking the time to read our response. We are glad the updated version addressed your concerns!

---

### Review · Reviewer_UJfx · 2024-10-31

**Summary Of Contributions:**

This work addresses a key issue in OOD detection by introducing INK, a novel OOD scoring approach that provides a rigorous likelihood-based interpretation for discriminative classifiers. INK operates on constrained latent embeddings modeled as a mixture of hyperspherical embeddings with constant norms, bridging hyperspherical distributions with intrinsic likelihood. Experiments show that INK achieves state-of-the-art performance across various OOD scenarios.

**Audience:**

Yes

**Broader Impact Concerns:**

I have no concern since there is enough discussion in the "Broader Impacts" section in the submission.

**Claims And Evidence:**

Yes

**Requested Changes:**

Major:
- Please provide more evidence and support for the uniform class prior assumption in Theorem 3.1, especially in the context of OOD detection. The authors can also address my concern by discussing why the proposed method is equivalent in principle to the previous method but achieves better performance in practice.
- Include a more thorough breakdown of why specific methods perform as they do, perhaps through additional ablation studies or statistical tests. More detailed discussions of the results can provide stronger evidence and help justify the effectiveness of the proposed method.

Clarity:
- What is the motivation of introducing "log" in Definition 3.2 (Intrinsic Likelihood Score)? Mathematically speaking, if there is no "log", the derivation of (8) changes from "+" to "×"?
- The authors may combine Section 5 with some discussion in Section 3.1? From my perspective of view, the content in Section 5 can be broken apart and incorporated in previous sections.

Typos:
- bold $\theta$ in equation (14)?

**Strengths And Weaknesses:**

Strengths
- Inspired by vMF distribution in the hypersphere, the authors provide Intrinsic Likelihood Score for OOD detection with theoretical analysis.
- They discuss the optimization issue of their method in DNN and class-Imbalanced datasets in practice.
- The extensive experiments impress me, and the proposed method achieves state-of-the-art performance across various OOD scenarios, which is promising.

Weaknesses
- The assumption in Theorem 3.1 appears quite strong, as it requires a uniform class prior. In practice, datasets are often imbalanced, so relaxing this assumption would render the derivation for Theorem 3.1 inapplicable, making the theorem’s applicability somewhat restricted.
- While the authors discuss "Class-Imbalanced Datasets" in Section 5, they show that their metric "maintains functional equivalence to the log-likelihood for OOD detection." So if your method is equivalent in principle to the previous method, why is your method better in practice?
- I'm not sure why the proposed method performs better than existing methods. Does it requires significant hyper-parameter tuning?

---

> ### Author Response · Authors · 2024-11-12
> **Response to Reviewer UJfx**
>
> Thank you for your thoughtful and detailed review of our paper. We appreciate the reviewer's recognition of our paper's clarity and extensive experiments.  We address each of your points below.
>
> > W1. Assumption in Theorem 3.1 on uniform class prior
>
> While Theorem 3.1 is derived under the assumption of a uniform class prior, our method can be extended to handle imbalanced datasets. We show that the INK score still performs well on imbalanced data by using empirical class frequencies or introducing weighted class probabilities. **We have relaxed this assumption in Section 5, showing that Theorem 3.1 still holds without such assumption**.
>
> > W2. _"they show that their metric "maintains functional equivalence to the log-likelihood for OOD detection." So if your method is equivalent in principle to the previous method, why is your method better in practice?"_
>
> Thank you for raising this question. It appears there was a misunderstanding in our claim. **We are not claiming our method is equivalent to previous methods**. _Instead, our theorem aims to establish equivalence between our INK score and **the ideal OOD detector**, as defined in Definition 2.2_. Our approach offers a distinct advantage by achieving a principled likelihood-based interpretation that previous heuristic-driven methods lack.
>
>
> Theorem 3.1 demonstrates that the INK score $S(z)$ is functionally equivalent to the log-likelihood $\log p(z)$ (up to a constant), achieving the theoretical properties expected of an ideal likelihood-based OOD detector. In contrast, prior OOD detection methods, including state-of-the-art approaches [1, 2, 3] for discriminative classifiers, either lack a rigorous log-likelihood interpretation or rely on strong, often impractical, assumptions about the density function.
>
>
> Our method approximates the ideal OOD detection criterion through a principled likelihood-based approach, bridging the gap between theoretical rigor and practical performance in OOD detection. The key distinction is that our approach achieves the equivalence to log-likelihood by explicitly modeling representations in constrained hyperspherical space. This provides a well-defined probability space where distances between embeddings have meaningful probabilistic interpretations. This allows us to draw a novel connection between the hyperspherical distribution and the density function under our probabilistic model.
>
> > W3. _"I'm not sure why the proposed method performs better than existing methods. Does it require significant hyper-parameter tuning?"_
>
> Thank you for your question. The proposed method performs better than existing methods primarily because it explicitly models class-conditional densities in a constrained hyperspherical space using the von Mises-Fisher (vMF) distribution. This principled approach aligns the training and test-time objectives, providing a consistent probabilistic interpretation that enhances OOD detection accuracy.
>
> In terms of hyperparameter tuning, our method does not require significant additional tuning compared to baseline methods. Key hyperparameters, such as the concentration parameter for the vMF distribution, are set based on standard validation procedures, and we found that the method is relatively robust across different datasets and OOD detection settings. **Overall, the gains in performance stem from the principled design of the model rather than extensive hyperparameter adjustments.**

---

> > ### Author Response · Authors · 2024-11-12
> > **Response to Reviewer UJfx (cont.)**
> >
> > > Supporting evidence for INK's effectiveness
> >
> > We appreciate your point regarding supporting evidence. To demonstrate the effectiveness of our method:
> >
> > - **we conducted an ablation study on the effect of hyperspherical embedding**. We investigate the impact of the hyperspherical embeddings by comparing performance with our training objective vs. standard cross-entropy loss. Figure 2 highlights the role of hyperspherical embeddings in achieving reliable OOD detection. We discuss the relationship of INK with prior work by [4]. The key distinction between the two methods lies in the geometry of the embedding space, from which the score is derived.
> > - **we conducted an ablation study on the effect of the OOD score function**, where we substituted the intrinsic likelihood score with alternative scoring functions such as non-parametric KNN distance. The contrast between INK vs. CIDER in Table 2 precisely demonstrates that the intrinsic likelihood score not only aligns well with our theoretical framework but also enhances empirical performance.
> > - **we conducted evaluation across diverse datasets and architectures**. We provide detailed experimental results across multiple datasets and OOD detection settings to showcase the robustness of our method. This includes performance comparisons on both near-OOD and far-OOD benchmarks, which reveal where our approach outperforms baselines. Discussions of various model architectures and datasets in **Appendices E and F**.
> > - **we included statistical significance tests**. We included a confidence interval for INK performance in Table 2. This analysis helps confirm that the improvements we observe are not due to chance but are statistically meaningful.
> > - **we included an embedding visualization analysis in Appendix G**, showing that our objective indeed results in compact vMF distributions.
> > - In response to your concern, we have added additional experiments in **Appendix I** of the revised manuscript to evaluate the INK score's effectiveness when classifiers are trained with standard cross-entropy or contrastive loss. As shown in Table 8, the model trained with vMF loss achieves a more robust hyperspherical embedding space, which enhances the INK score’s effectiveness compared to models trained with other loss functions, aligning well with the log-likelihood representation.
> >
> > > Clarity: Log term in Definition 3.2
> >
> > The primary motivation for introducing the "log" in Definition 3.2 (Intrinsic Likelihood Score) is to align with existing log-likelihood interpretations commonly used in probabilistic models. This alignment ensures consistency with traditional likelihood-based methods, making the score more interpretable in the context of OOD detection.
> >
> >
> > > Clarity: Combining Section 5 and Section 3.1
> >
> > Thank you for your suggestion! We have revised the manuscript to include the extended INK derivation addressing class imbalance in Section 3.1, with the results now presented in Section 4.3.
> >
> > > Typos
> >
> > All fixed – thank you for the careful read!
> >
> > [1] Djurisic et al. Extremely Simple Activation Shaping for Out-of-Distribution Detection. ICLR 2023
> > [2] Ming et al. How to Exploit Hyperspherical Embeddings for Out-of-Distribution Detection? ICLR 2023
> > [3] Sun et al. Out-of-Distribution Detection with Deep Nearest Neighbors. ICML 2022
> > [4] Liu et al. Energy-based out-of-distribution detection. NeurIPS 2020

---

> > > ### Comment · Reviewer_UJfx · 2024-11-12
> > > **All my concerns have been addressed**
> > >
> > > My major concerns have been addressed, the updated version is more clear.

---

> > > > ### Author Response · Authors · 2024-11-12
> > > > **follow up**
> > > >
> > > > Thank you for taking the time to read our response. We are glad the updated version addressed your concerns!

---

### Review · Reviewer_w67q · 2024-10-31

**Summary Of Contributions:**

This paper proposes a novel framework called INtrinsic liKelihood (INK) for out-of-distribution (OOD) detection in classification tasks. Through extensive experiments, the authors show that INK provides competitive performance compared to the state-of-the-art methods on different datasets including ImageNet-1K and CIFAR-100 as well as multiple architectures such as ResNet and Vision Transformer (ViT).

**Audience:**

Yes

**Claims And Evidence:**

Yes

**Requested Changes:**

1. The paper argues that the problems with previous works are: (I) "heuristic-driven and cannot be rigorously interpreted as log-likelihood", and (II) "impose strong assumptions on the density function that can fail to hold in practice".
The authors should explain why inability to interpret a method as log-likelihood is necessary a limitation for a given method. Moreover, they should clarify whether INK suffers from problem (II), i.e. strong assumptions on the density function.

2. The authors should explain why their method on CIFAR-100 and ViT is less efficient compared to ImageNet-1K and ResNets. Does it relate to the number of classes of the dataset, the resolution of the images, the convolutional layer vs. linear layer, the normalization layer (BatchNorm in ResNets and LayerNorm in ViTs), and etc?

3. Based on (2), the authors should revise some claims in the paper such as "As shown in Table 7, our INK score maintains competitive performance with the ViT model". To me, INK underperforms the existing methods in many cases for the ViT architecture and CIFAR-100.

**Strengths And Weaknesses:**

**Strengths**:

The paper
1. is well-written and well-presented.
2. provides theoretical insights.
3. provides extensive experiments to compare the proposed method with the state-of-the-art approaches.
4. honestly discusses the limitations of INK.

**Weaknesses**:
1. The model needs to be trained under the vMF distribution, which can limit the usage of INK in practice.
2. According to the results in Table 6, it seems that unlike ImageNet-1k, INK is less efficient on CIFAR-100 compared to the state-of-the-art methods. Similar observation holds for ViT (Table 7).

---

> ### Author Response · Authors · 2024-11-12
> **Response to Reviewer w67q**
>
> Thank you for your thoughtful and detailed review of our paper. We appreciate your recognition of our contributions, particularly in theoretical insights and empirical rigor. We address each of your points below.
>
>
> > W1. The model needs to be trained under the vMF distribution, which can limit the usage of INK in practice.
>
> Thank you for highlighting this. Indeed, we have acknowledged this in our limitation section:
>
> _"A key strength of our method is its grounded interpretation from a likelihood perspective. This approach, however, necessitates training the model under the vMF distribution. While our method achieves accuracy comparable to state-of-the-art models in certain scenarios, it's important to note that it is less common than models trained using traditional cross-entropy loss. This may limit its immediate applicability in environments where standard cross-entropy loss models are the norm. However, we believe that the trustworthiness of the models is worth the effort of training them in a certain way."_
>
>
> > Limitation of not interpreting as log-likelihood
>
> This is an insightful point. The absence of a log-likelihood interpretation in some OOD methods can indeed be a drawback. Discriminative classifiers generally optimize for $p(y|z)$ (posterior), not $p(z)$ (data likelihood), often through softmax cross-entropy, i.e., $p(y|z) = \frac{e^{f_y(z)}}{\sum_{i=1}^C e^{f_i(z)}}$. Without an explicit model for $p(z)$, these approaches lack a principled probabilistic interpretation of whether data belongs to the in-distribution, as defined in Definition 2.2. Our likelihood-based approach provides this probabilistic grounding, enabling stronger and more reliable OOD detection.
>
>
> > Whether INK suffers from problem (II), i.e., strong assumption on the density function
>
> We would like to clarify that our original statement intends to highlight previous methods such as Mahalanobis distance which impose strong assumptions on the feature embedding space, when the features are not explicitly trained to form such density. In other words, there is a mismatch between the actual embedding distribution vs. the assumption used in the test-time score.
>
> In contrast, INK explicitly shapes the representations into the desired distribution during training, rather than assuming some distribution post hoc. Our approach yields an OOD score where the density function is mathematically consistent during training and testing, avoiding the mismatch in the previous method. Specifically, INK explicitly models class-conditional densities using the von Mises-Fisher (vMF) distribution on a hyperspherical space. INK’s scoring function, or intrinsic likelihood, relates directly to the log-likelihood $\log p(z)$ (up to a constant). INK thus bridges the gap by enabling principled likelihood estimation within a classification-based model. The key insight is that we need to bring explicit probabilistic modeling of $p(z)$ during the optimization of $p(y|z)$, which is effectively achieved by our framework.
>
> > Clarification on CIFAR-100 and ViT performance
>
> Thank you for noting this. On CIFAR-100, _INK shows strong performance on the far-OOD benchmark (average FPR 44.51%), outperforming state-of-the-art methods for large distribution shifts_. Performance on near-OOD, however, is comparable across methods, likely due to CIFAR’s lower resolution compared to ImageNet-1k. For ViT, while INK achieves competitive results, it does not consistently outperform all methods. We will clarify these nuances in Section 4.2 to highlight both strengths and trade-offs in INK’s performance across datasets and OOD types.
>
> Thank you again for your valuable feedback.

---

> ### Comment · Reviewer_w67q · 2024-11-27
> **Major concerns addressed.**
>
> The authors addressed my major concerns.

---

> > ### Author Response · Authors · 2024-11-27
> >
> > Thank you for taking the time to read our response. We are glad the updated version addressed your concerns!

---

### Author Response · Authors · 2024-11-12
**Summary of response – Thank you all reviewers for your insightful feedback!**

Thank you all the reviewers for their constructive and valuable feedback. We are glabd that reviewers find our work supported by **theoretical insights** (R1, R2, R3) with **extensive experiments** (R1, R2, R3) and shows promising **performance and efficiency** (R3). We are encouraged that reviewer found the paper **well-written** (R1).

We have addressed the reviewers' comments and concerns in individual responses to each reviewer. The reviews give us the opportunity to revise our manuscript. We have carefully addressed the requested changes and made several improvements. Changes and additions are highlighted in blue in the revised manuscript, with major updates including:

* [R1] Highlighted the key insight on bringing explicit probabilistic modeling of $p(z)$ during the optimization of $p(y|z)$.
* [R1] Improved the clarity of discussion on CIFAR and ViT ablation studies in Appendix E and F
* [R2] Introduce class imbalance in Section 3.1, with the results in Section 4.3.
* [R2] Fixed the typos.
* [R2, R3] Added more ablation studies in Appendix I, evaluating INK's effectiveness with different objectives.

Thanks for the careful read!

(* As abbreviations, we refer to **Reviewer w67q** as R1, **Reviewer UJfx** as R2, and **Reviewer nD6V** as R3, respectively)

---

### Decision · Action_Editor_1LRw · 2024-11-29

**Recommendation:** Accept as is

**Comment:**

This paper proposed a novel OOD detection method, called INtrinsic liKelihood (INK). This is inspired by the fact that classifiers are implicitly trained to maximize the likelihood of embeddings matching in distribution (ID) data. Extensive empirical results verify the effectiveness of the proposed detection method.

**Decision** All the reviewers appreciate the technical and experimental contributions. They all recommended the acceptance, I quickly went through the paper and agreed with their recommendations.

**Audience:**

Yes.

**Claims And Evidence:**

Yes.